

# Cranial ornamentation in the Late Cretaceous nodosaurid ankylosaur *Hungarosaurus*

Attila Ősi[1,3], János Magyar[1], Károly Rosta[1] and Matthew Vickaryous[2]

[1] Department of Paleontology, Eötvös Loránd University, Budapest, Hungary
[2] Department of Biomedical Sciences, University of Guelph, Guelph, Canada
[3] Hungarian Natural History Museum, Budapest, Hungary

## ABSTRACT

Bony cranial ornamentation is developed by many groups of vertebrates, including ankylosaur dinosaurs. To date, the morphology and ontogenetic origin of ankylosaurian cranial ornamentation has primarily focused on a limited number of species from only one of the two major lineages, Ankylosauridae. For members of the sister group Nodosauridae, less is known. Here, we provide new details of the cranial anatomy of the nodosaurid *Hungarosaurus* from the Santonian of Europe. Based on a number of previously described and newly identified fragmentary skulls and skull elements, we recognize three different size classes of *Hungarosaurus*. We interpret these size classes as representing different stages of ontogeny. Cranial ornamentation is already well-developed in the earliest ontogenetic stage represented herein, suggesting that the presence of outgrowths may have played a role in intra- and interspecific recognition. We find no evidence that cranial ornamentation in *Hungarosaurus* involves the contribution of coossified osteoderms. Instead, available evidence indicates that cranial ornamentation forms as a result of the elaboration of individual elements. Although individual differences and sexual dimorphism cannot be excluded, the observed variation in *Hungarosaurus* cranial ornamentation appears to be associated with ontogeny.

## INTRODUCTION

Development of osseous cranial ornamentation is a relatively common occurence in the evolutionary history of terrestrial vertebrates (*De Buffrénil, 1982*). Among reptiles, cranial ornamentation, including frills, crests, horns, bosses, or casques, is known for representative members of many fossil and extant groups (e.g., *Gadow, 1901*; *Romer, 1956*; *Clarac et al., 2017*; *Mayr, 2018*). The ultimate morphology of cranial ornamentation, especially among skeletally mature adults, is often highly variable and species-specific (e.g., *Otto, 1909*; *Montanucci, 1987*). As currently understood, this vast diversity is the result of two principal modes of morphogenesis: (1) the fusion of additional skeletal

Corresponding author
Attila Ősi, hungaros@gmail.com

elements, commonly identified as osteoderms, with the skull; and (2) the elaboration of individual cranial elements (*Moss, 1969*; *Vickaryous, Russell & Currie, 2001*).

Osteoderms (= dermal sclerifications, osteoscutes) are bone-rich elements that form within the dermis of the skin (*Moss, 1969*; *Vickaryous & Sire, 2009*). As demonstrated by modern lizards, osteoderms that develop across the head contribute to the formation of a highly variable polygonal-like pattern of cranial ornamentation that embosses the superficial surface of the skull and mandible (Figs. 1A–1D). The extent to which osteoderm contact or even fuse with the skull is both species-specific and ontogenetically variable (*Vickaryous, Russell & Currie, 2001*; *Bhullar, 2011*; *Paluh, Griffing & Bauer, 2017*; *Maisano et al., 2019*; *Laver et al., 2020*). While in some species, osteoderms always remain suspended within the dermis itself (e.g., some gekkotan lizards; *Paluh, Griffing & Bauer, 2017*; *Laver et al., 2020*), in other taxa they gradually fuse with subadjacent bones of the skull (e.g., helodermatids, xenosaurids; *Bhullar, 2011*; *Maisano et al., 2019*). As osteoderms develop within the skin, their development is not restricted to the area of an individual bone, and hence they routinely occupy positions that overlap sutural boundaries (*Vickaryous, Russell & Currie, 2001*).

In addition to the fusion of osteoderms, cranial ornamentation may also develop as a result of the elaboration or exaggerated outgrowth of individual cranial (and mandibular) elements (Figs. 1E–1H) (e.g., *Montanucci, 1987*; *Vickaryous, Russell & Currie, 2001*; *Hieronymus et al., 2009*). In some species, particularly among aged individuals, this form of exaggerated outgrowth may become continuous across multiple adjacent bones (e.g., "hummocky rugosities"; *Hieronymus et al., 2009*).

Cranial ornamentation is one of the most diagnostic features of the extinct archosaur clade Ankylosauria (*Maryańska, 1977*; *Coombs Jr, 1978*; *Carpenter et al., 2001*; *Vickaryous, Maryańska & Weishampel, 2004*). For most ankylosaur taxa, the dorsolateral surfaces of the cranium and the posterolateral surface of the mandible are externally (superficially) embossed with cranial ornamentation. Although intraspecific (and possibly ontogenetic) variation exists, details of the size, shape and pattern of cranial ornamentation, often referred to as 'caputegulae' (*Blows, 2001*), have long been recognized as taxonomically informative (e.g., *Parks, 1924*; *Coombs Jr, 1971*; *Coombs Jr, 1978*; *Blows, 2001*; *Penkalski, 2001*; *Arbour & Currie, 2013*; *Arbour & Currie, 2016*). This includes the classical distinction of the two major clades of ankylosaurs: Ankylosauridae and Nodosauridae (*Coombs Jr, 1978*).

The ontogenetic origin of cranial ornamentation in ankylosaurs has primarily focused on a handful of species (*Leahey et al., 2015*), most of which are members of Ankylosauridae (*Coombs Jr, 1971*; *Vickaryous, Russell & Currie, 2001*; *Carpenter et al., 2001*; *Hill, Witmer & Norell, 2003*). Based on the investigation of multiple specimens of the ankylosaurids *Euoplocephalus* and *Pinacosaurus*, including material attributed to subadult (i.e., not skeletally mature) individuals, the cranial ornamentation of these forms are interpreted involving both the coosification of osteoderms with the skull and the exaggerated outgrowth of individual cranial elements (*Vickaryous, Russell & Currie, 2001*; *Hill, Witmer & Norell, 2003*; although see *Carpenter et al., 2001*). A similar combination of processes has been proposed for the basal ankylosaurian *Kunbarrasaurus ieversi* (*Leahey et al., 2015*). In

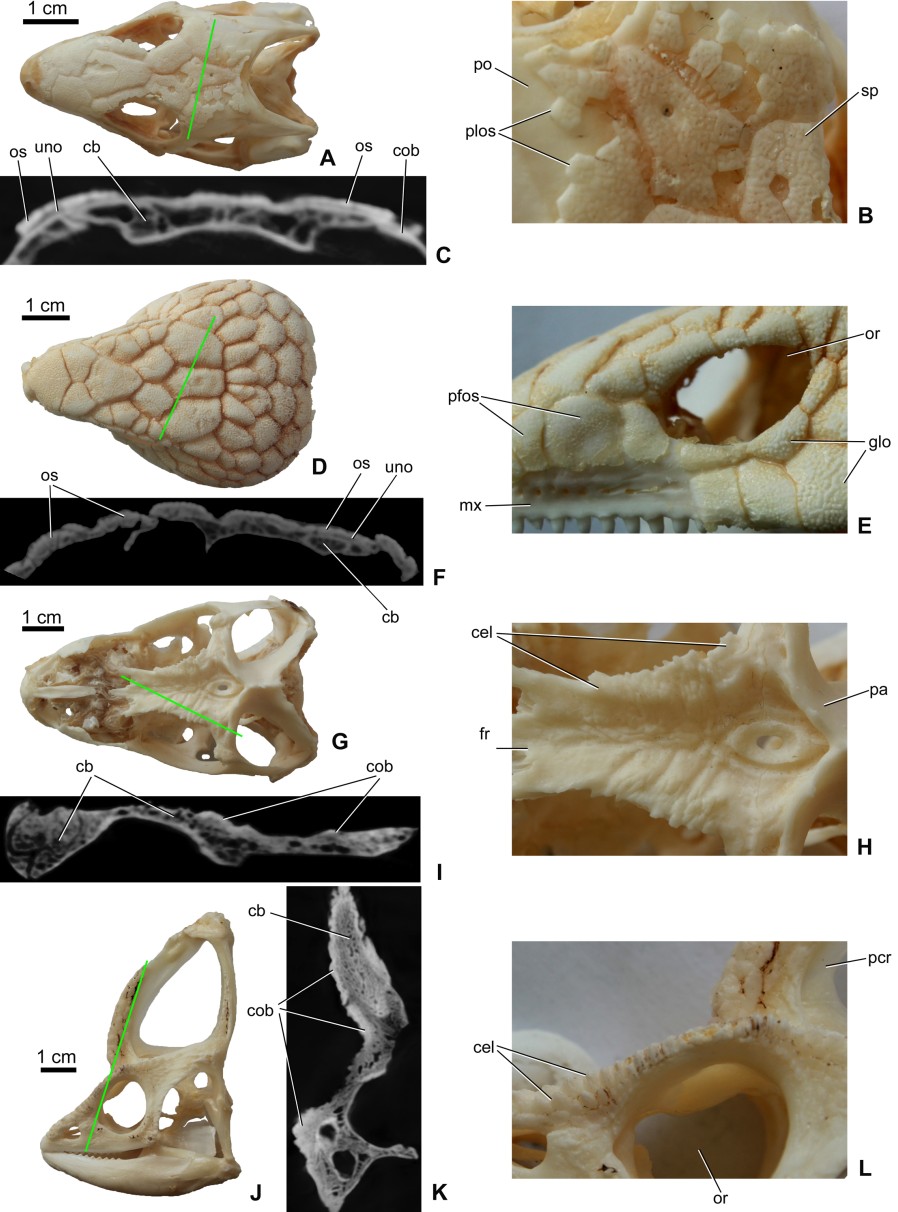

**Figure 1** Surface view and microCT cross sectional images (in level of the green line) of cranial ornamentation developed as either osteodermal fusion (A–F) or elaboration of skull bones (G–L) in squamates. (A) *Tiliqua scincoides* skull (MDE R45) in dorsal view. (B) Partially fused polygonal osteoderms on the skull of *T. scincoides*. (C) Inner structure of the skull bones and covering osteoderms of *T. scincoides*. (D) *Tiliqua nigrolutea* skull (MDE R47) in dorsal view. (E) Partially fused polygonal osteoderms on the skull of *T. nigrolutea*. (F) Inner structure of the skull bones and covering osteoderms of *T. nigrolutea*. (G) *Iguana iguana* skull (MDE R20) in dorsal view. (H) Elaboration of skull bones in *I. iguana*. (I) Inner structure of the elaborated skull bones in *I. iguana*. (J) *Chamaeleo calyptratus* (MDE R43) skull in lateral view. (K) Inner structure of the elaborated skull bones in *C. calyptratus*. (L) Elaboration of skull bones in *C. calyptratus*. Abbreviations: cb, cancellous bone; cel, cranial elaboration; cob, compact bone; fr, frontal; glo, globular ornamentation; mx, maxilla; or, orbit, os, osteoderm; pa, parietal; pcr, parietal crest; pfos, partially fused osteoderms; plos, polygonal osteoderms; po, postorbital; uno, unossified part between osteoderm and skull bone; sp, small pits.

contrast, osteoderms do not appear to fuse with the skulls of some basal taxa *Cedarpelta* (*Carpenter et al., 2001*) and *Gastonia* (*Kinneer, Carpenter & Shaw, 2016*). Hence, cranial ornamentation in these species appears to be exclusively the result of elaborated outgrowth of individual elements. Among nodosaurids, less is known. Although a partial skull (attributed to an unidentified species) was reported to demonstrate a rugose external texture, with no evidence of "... overgrowth of dermal bone" (*Jacobs et al., 1994*), the specimen is fragmentary, incomplete, and skeletally immature. Therefore, the developmental processes involved in the formation of cranial ornamentation among nodosaurids remains uncertain.

## Cranial ornamentation in extant amniotes

Many extant groups of non-iguanian lizards develop osteoderms across the dorsal and lateral surfaces of the skull (Table 1; see also *Gadow, 1901*; *Moss, 1969*; *Montanucci, 1987*; *Etheridge & De Queiroz, 1988*; *Vickaryous & Sire, 2009*). Although the morphology and arrangement of osteoderms across the skull is taxonomically variable (e.g., Figs. 1A–1D; see also *Mead et al., 2012*; *Ledesma & Scarpetta, 2018*), evidence for sexual dimorphism remains limited (Table 1). For most species, both males and females develop comparable arrangements of osteoderm-based ornamentation (see references in Table 1). One possible exception is the skeletally mature marine iguana (*Amblyrhynchus cristatus*). Marine iguanas are one of the only iguanid lizards that have been identified as developing osteoderms, and these elements only form on the head (*Etheridge & De Queiroz, 1988*). In females, cranial ornamentation is reportedly less developed than that of males (*Eibl-Eibesfeldt, 1966*).

Unlike lizards, osteoderms are absent from the heads of modern archosaurs (birds and crocodylians), with the possible exception of the bony crocodylian palpebral (eyelid bone) (*Vickaryous & Hall, 2008*).

Whereas cranial ornamentation in many non-iguanians is characterized by osteoderms, that of iguanians is dominated by the elaboration and outgrowth of individual skull (and specifically dermatocranial) elements (*Etheridge & De Queiroz, 1988*; see Table 1). This outgrowth form of cranial ornamentation primarily manifests as rugosities with variably developed crests, pits and bumps (*Hieronymus et al., 2009*, Figs. 1E–1F), although some taxa may develop large horn-like structures as well. For example, in species of *Phrynosoma* horns and bosses can develop on both the parietal and squamosal (*Lang, 1989*; *Vickaryous, Russell & Currie, 2001*; *Powell et al., 2017*). Although the number, morphology and orientation of these protuberances can vary among *Phrynosoma* species, there is no evidence that they are sexual dimorphic (*Powell et al., 2017*, see Table 1). Similarly, anoles (Dactyloidae) also develop taxon-specific cranial ornamentation that is present in both sexes (*Etheridge & De Queiroz, 1988*).

The exaggerated development of bony horns and crests is also characteristic of many archosaurs, including fossil (e.g., *Ceratosuchus Schmidt, 1938*; *Bartels, 1984*; *Brochu, 2006*; *Brochu, 2007*; *Bickelmann & Klein, 2009*) and extant (e.g., *Crocodylus rhombifer*; *Brochu et al., 2010*) crocodylians. Among modern crocodylains, these protuberances are not sexually dimorphic, but may be used for species recognition in ecosystems where multiple taxa of crocodylians exist (*Bartels, 1984*). Cranial ornamentation is also characteristic of many

Ösi et al. (2021), *PeerJ*, DOI 10.7717/peerj.11010

**Table 1 Osseous cranial ornamentation in extant sauropsid vertebrates.**

| Development of cranial ornamentation | Sexual variation | Higher-level taxon | Family | *Genus/species* example | Morphology | Function | Reference |
|---|---|---|---|---|---|---|---|
| Osteoderms | Monomorphic | Squamates | Helodermatidae | *Heloderma* | Flat, scale-like | ? | *Mead et al. (2012)* |
| Osteoderms | Monomorphic | Squamates | Gerrhonotide | *Abronia, Barisia, Mesaspis* | Flat, scale-like | ? | *Ledesma & Scarpetta (2018)* |
| Osteoderms | Monomorphic | Squamates | Gerrhosauridae | *Angolosaurus, Tracheloptychus* | Flat, scale-like or harply keeled scales | ? | *Nance (2007)* |
| Osteoderms | Monomorphic | Squamates | Scincidae | *Eugongylus, Eumeces, Tiliqua* | Flat, polygonal | ? | *Čerňanský & Hutchinson (2013)* |
| Osteoderms | Monomorphic | Squamates | Xenosauridae | *Xenosaurus* | Flat, scale-like | ? | *Smith, Lemos-Espinal & Ballinger (1997)* |
| Osteoderms | Monomorphic | Squamates | Xantusiidae | *Lepidophyma gaigeae* | Flat, scale-like | Physical protection? | *Peterson & Bezy (1985)* and *Ramírez-Bautista et al. (2008)* |
| Osteoderms | Monomorphic | Squamates | Cordylidae | *Ouroborus, Cordylus cataphractus* | Flat or slightly domed, pointed | Intrasexual fight | *Broeckhoven, De Kock & Mouton (2017)*, *Broeckhoven, Du Plessis & Hui (2017)*, *Broeckhoven, DeKock & Hui (2018)* and *Flemming, Bates & Broeckhoven (2018)* |
| Osteoderms | Monomorphic | Squamates | Lanthanotidae | *Lanthanotus borneensis* | Small, flat to convex | ? | *Maisano et al. (2002)* |
| Osteoderms | Monomorphic | Squamates | Lacertidae | *Lacerta strigata, Xantusia riversiana* | Flat, scale-like | ? | *Čerňanský & Syromyatnikova (2019)* |
| Osteoderms | Monomorphic | Squamates | Gekkonidae | *Gekko gecko* | Flat, scale-like | ? | *Laver et al. (2020)* |
| Osteoderms | Monomorphic | Squamates | Phyllodactylidae | *Tarentola mauritanica* | Flat, scale-like | ? | *Paluh, Griffing & Bauer (2017)* and *Levrat-Calviac & Zylberberg (1986)* |
| Osteoderms | Monomorphic | Squamates | Varanidae | *Varanus komodoensis* | Small, vermiform osteoderms | ? | *Maisano et al. (2019)* and *Kirby et al. (2020)* |
| Osteoderms | Dimorphic | Squamates | Iguanidae | *Amblyrhynchus* | Knob-like | Interlock the horns during breeding | *Eibl-Eibesfeldt (1966)* and *Wikelski & Trillmich (1997)* |
| Skull elaboration | Monomorphic | Squamates | Phrynosomatidae | *Phrynosoma* | High spikes | Interspecific | *Montanucci (1987)* and *Powell et al. (2017)* |
| Skull elaboration | Monomorphic | Squamates | Dactyloidae | *Anolis* spp. | Hummocky rugosity, small crests | Interspecific | *Etheridge & De Queroz (1988)* and *Tinius (2019)* |
| Skull elaboration | Monomorphic | Squamates | Carphodactylidae | *Phyllurus cornutus* | Hummocky rugosity | ? | *Doughty & Shine (1995)* |
| Skull elaboration | Monomorphic | Squamates | Teiidae | *Cnemidophorus lemniscatus* | Hummocky rugosity, small crests | ? | *Anderson & Vitt (1990)* |
| Skull elaboration | Monomorphic | Squamates | Corytophanidae | *Corytophanes* | Casque, crest | ? | *Lang (1989)*, *Taylor et al. (2017)* and *Smith (2011)* |
| Skull elaboration | Monomorphic | Turtles | Chelidae | *Chelus fimbriata* | Shallow hummocky rugosity | ? | *Garbin & Caramaschi (2015)* |
| Skull elaboration | Monomorphic | Turtles | Chelydridae | *Macrochelys temminckii* | Hummocky rugosity, small grooves | ? | – |

*(continued on next page)*

Ősi et al. (2021), PeerJ, DOI 10.7717/peerj.11010

**Table 1** (*continued*)

| Development of cranial ornamentation | Sexual variation | Higher-level taxon | Family | *Genus/species* example | Morphology | Function | Reference |
|---|---|---|---|---|---|---|---|
| Skull elaboration | Monomorphic | Turtles | Testudinidae | *Geochelone denticulata* | Hummocky rugosity, small grooves | ? | *Gaffney (1979)* |
| Skull elaboration | Monomorphic | Crocodiles | Crocodylidae | *Crocodylus rhombifer* | Squamosal horn | Interspecific | *Bartels (1984)* and *Brochu et al. (2010)* |
| Skull elaboration | Monomorphic | Birds | Casuariidae | *Casuarius* spp. | Casque | Thermal radiator | *Naish & Perron (2016)* and *Eastick et al. (2019)* |
| Skull elaboration | Monomorphic | Birds | Bucorvidae | *Bucorvus* spp. | Frontal hump | Species recognition, amplify communication | *Alexander, Houston & Campbell (1994)* |
| Skull elaboration | Monomorphic | Birds | Numididae | *Numida meleagris* | Casque | Thermoregulation, vocalisation and intraspecific combat? | *Mayr (2018)* |
| Skull elaboration | Monomorphic | Birds | Megapodiidae | *Macrocephalon maleo* | Vauled skull | ? | *Green & Gignac (2019)* |
| Skull elaboration | Monomorphic | Birds | Gruidae | *Balearica regulorum* | Frontal hump, horn-like tubercles on parietal | ? | *Mayr (2018)* |
| Skull elaboration | Monomorphic | Birds | Anatidae | *Anas gibberifrons* | Frontal hump | Physiological, sensory, or acoustic function? | *Mayr (2018)* |
| Skull elaboration | Monomorphic | Birds | Cracidae | *Oreophasis derbianus, Pauxi* | Casque | Demonstrative of ability to survive | *Vaurie (1968)*, *González-García (1995)* and *Mayr (2018)* |
| Skull elaboration | Monomorphic | Birds | Alcidae | *Fratercula arctica, Cerorhinca monocerata* | Crest or horn on the upper beak | Beak strengthening? | *Jones (1993)* |
| Skull elaboration | Monomorphic | Birds | Pelecanidae | *Pelecanus erythrorhynchos* | Crest on the upper beak | Display during breading | *Evans & Knopf (1993)* |
| Skull elaboration | Monomorphic | Birds | Procellariidae | *Pagodroma nivea, Fulmarus glacialis* | Crest on the upper beak | ? | *Jouventin & Viot (1985)* |
| Skull elaboration | Monomorphic | Birds | Chionididae | *Chionis minor* | Shield-like callosity | Physiological role? | *Lowe (1916)* and *Mayr (2018)* |
| Skull elaboration | Monomorphic | Birds | Musophagidae | *Musophaga violacea* | Casque | ? | *Mayr (2018)* |
| Skull elaboration | Monomorphic | Birds | Icteridae | *Psarocolius decumanus* | Crest on the upper beak | ? | *Webster (1992)* and *Fraga & Kreft (2007)* |
| Skull elaboration | Monomorphic | Birds | Threskiornithidae | *Geronticus calvus* | | ? | *Kopij (1998)* |
| Skull elaboration | Monomorphic | Birds | Meliphagidae | *Philemon corniculatus* | Crest on the beak | ? | *Mayr (2018)* |

Ősi et al. (2021), *PeerJ*, DOI 10.7717/peerj.11010

**Table 1** (*continued*)

| Development of cranial ornamentation | Sexual variation | Higher-level taxon | Family | *Genus/species* example | Morphology | Function | Reference |
|---|---|---|---|---|---|---|---|
| Skull elaboration | Monomorphic | Birds | Cuculidae | *Crotophaga sulcirostris* | Deep upper beak | ? | *Posso & Donatelli (2001)* and *Mayr (2018)* |
| Skull elaboration | Dimorphic | Squamates | Corytophanidae | *Basiliscus* | Casque, crest | Intersexual | *Lang (1989)*, *Taylor et al. (2017)* and *Smith (2011)* |
| Skull elaboration | Dimorphic | Squamates | Chamaeleonidae | *Chameleo jacksoni, Triceros* | Crest, horns | Social significance, species recognition | *Rand (1961)* and *Eckhardt et al. (2012)* |
| Skull elaboration | Dimorphic | Birds | Phasianidae | *Tetrao urogallus* | Preorbital ridge | ? | *Lindén & Väisänen (1986)* |
| Skull elaboration | Dimorphic | Birds | Anatidae | *Cygnus, Melanitta, Oxyura* | Frontal hump | Fat reservoir indicating individual fitness | *Horrocks, Perrins & Charmantier (2009)*, *Lüps (1990)* and *Mayr (2018)* |
| Skull elaboration | Dimorphic | Birds | Anseranatidae | *Anseranas* | Frontal hump | ? | *Mayr (2018)* |
| Skull elaboration | Dimorphic | Birds | Cracidae | *Crax* | casque | Demonstrative of ability to survive | *Buchholz (1991)* and *Mayr (2018)* |
| Skull elaboration | Dimorphic | Birds | Bucerotidae | *Rhyticeros* | Casque on upper beak | ? | *Kemp (2001)* and *Mayr (2018)* |
| Osteoderms and skull elaboration | Monomorphic | Squamates | Anguidae | *Pseudopus (Ophisaurus) apodus* | Flat, scale-like, pitted osteoderms; grooved nasal, frontal, parietal | ? | *Klembara et al. (2017)* |

taxa of birds (Table 1). In most cases these elaborations and outgrowths are monomorphic (*Mayr, 2018*). One of the most obvious examples are cassowaries (*Casuarius* spp.), where males and females are similarly ornamented with elaborate casques on the skull roof (*Naish & Perron, 2016*). The internal bony architecture of this cranial ornamentation can also vary. For example, the casque on the upper bills of bucorvid and some bucerotid birds is typically dominated by an air-filled cavity and thin trabecular bone, but is reportedly solid bone in the greater helmeted hornbill (*Buceros vigil*) (*Gamble, 2007*).

Here we describe several fragmentary skulls and skull elements of the European Late Cretaceous (Santonian) nodosaurid ankylosaur, *Hungarosaurus* (Table 2). These specimens represent at least three different size classes (and likely different stages of ontogeny), and provide new information about the morphological diversity, development and possible function of cranial ornamentation of nodosaurid skulls. We compared our findings with gross anatomical and micro-computed tomography (microCT) data from the study of cranial ornamentation in modern lizards.

## MATERIAL AND METHODS

### Specimens

The Hungarian nodosaurid ankylosaur specimens used in this study (Table 2) are from the Upper Cretaceous (Santonian) Csehbánya Formation of the Iharkút vertebrate site, Bakony Mountains, western Hungary (*Ősi et al., 2019*; for geology and taphonomy, see *Botfalvai, Ősi & Mindszenty, 2015*; *Botfalvai et al., 2016*). Four partial nodosaurid ankylosaur skulls (Fig. 2) and various isolated skull elements (see Table 2 for all used specimens) from Iharkút are briefly described and compared in detail particularly focusing on the morphology, topographic distribution and origin of the cranial ornamentations. Two of the fragmentary skulls (holotype, MTM PAL 2013.23.1., Figs. 2A and 2D) and some isolated elements have been already described in more detail (*Ősi, 2005*; *Ősi & Makádi, 2009*; *Ősi, Pereda-Suberbiola & Földes, 2014*; *Ősi et al., 2019*), but cranial ornamentation was not discussed. The two new partial skulls (MTM PAL 2020.31.1., MTM PAL 2020.32.1., Figs. 2B, 2C, 2D and Data S1) have not been described in detail, and the comparative osteological description of these specimens are in Data S1.

The basis of this work is that all four skulls and isolated remains are thought to belong to *Hungarosaurus*. Although the presence of the much smaller *Struthiosaurus* at the site has also been confirmed by postcranial findings (*Ősi & Prondvai, 2013*; *Ősi & Pereda-Suberbiola, 2017*), the two new skulls are closer to *Hungarosaurus* based on the osteological features listed in Data S1. The postorbital crest of the specimen MTM PAL 2020.32.1. is, however, somewhat different from that of the holotype of *Hungarosaurus*.

In addition to the fossil specimens, we performed a comparative micro-computed tomography (microCT) investigation on one skull each of four extant lizard species: *Tiliqua scincoides* (MDE R45); *Tiliqua nigrolutea* (MDE R47); *Iguana iguana* (MDE R20); and *Chamaeleo calyptratus* (MDE R43).

**Table 2  List of *Hungarosaurus* specimens used in this study.**

| Reference No. | Preserved skull elements (elements with bold used in this study) | Ontogenetic stage | First described in: |
|---|---|---|---|
| Holotype, MTM 2007.26.1.-2007.26.34. | Partial skull including the **pre-maxillae**, **right postorbital and jugal**, ?left prefrontal, lacrimal, and **frontal**, posterior part of the pterygoid, both quadrates, condylus occipitalis, 22 teeth, one hyoid? bone, ?vomer, **anterior end of left nasal** | Adult | *Ősi (2005)*, *Ősi & Makádi (2009)*, *Ősi etal. (2019)* and *Botfalvai, Prondvai & Ősi (2020)* |
| MTM PAL 2013.23.1 | Skull fragment with **parietal** and basicranium | Adult? | *Ősi, Pereda-Suberbiola & Földes (2014)* |
| MTM PAL 2020.31.1. | Partial skull with most of the rostrum including the **premaxillae**, **nasals**, the right fragmentary maxilla and the right **frontal-supraorbital- ?prefrontal-?lacrimal** complex. | Subadult-to adult? | This study |
| MTM PAL 2020.32.1. | Partial skull including the partial basicranium, most of the **skull roof** (frontal, postfrontal, parietal) between and behind the orbits, the **two nasals**, the **left postorbital**, left squamosal, most of the left quadrate and the distal end of the right quadrate. | Subadult? | This study |
| MTM V.2003.12 | Isolated **left premaxilla** and partial maxilla | Juvenile | *Ősi & Makádi (2009)* |
| MTM PAL 2020.33.1. | Isolated **Left premaxilla** | Subadult? | This study |
| MTM V 2010.1.1. | Isolated **left postorbital and jugal** | Subadult? | *Ősi etal. (2012)* |
| MTM 2007.28.1. | Isolated **left postorbital** | subadult? | *Ősi & Makádi (2009)* |
| MTM 2007.27.1. | Isolated **left frontal** (originally described as nasal) | Subadult? to adult | *Ősi & Makádi (2009)* |
| MTM PAL 2020.34.1. | Isolated **right nasal** | adult? | This study |

## Methods

Specimens were collected between 2001 and 2019 and all of them are housed in the Vertebrate Paleontological Collection of the Hungarian Natural History Museum, Budapest (MTM). Specimens were prepared mechanically in the labs of the Department of Paleontology of the Eötvös University and the Hungarian Natural History Museum, and the bones were pieced together using cyanoacrylate glue.

For 3D reconstruction of the skulls (Fig. 2), we photographed each bone with a Canon EOS 600D DS126311 camera using photogrammetry. 2D images were converted to 3D images using open source 3DF Zephyr software (version 4.5.3.0). 3D images of bones also show the original surface texture of the bones. 3D files of each bone were assembled within the open source software Blender using Polygonal modeling and Sculpting techniques. Finally, we rendered a turntable video of the digitally finalized skull in Marmoset Toolbag

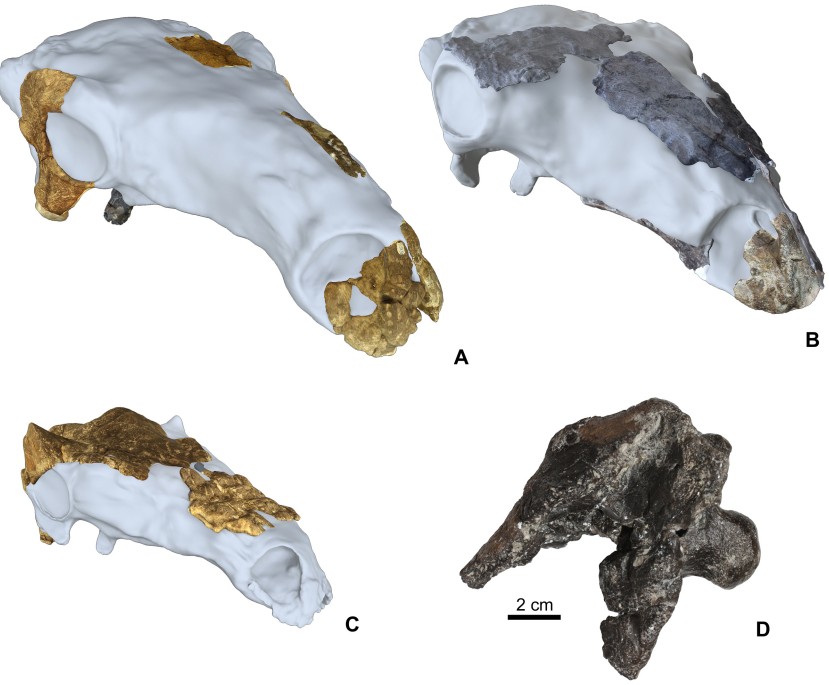

**Figure 2** **Cranial specimens of the Late Cretaceous (Santonian) nodosaurid ankylosaur,**
***Hungarosaurus tormai* in 3D reconstruction (for 3D reconstruction see Video S1–S6).** (A) holotype
skull (MTM 2007.26.1.-2007.26.34.). (B) MTM PAL 2020.31.1. (C) MTM PAL 2020.32.1. (D) basicranium
and partial skull roof MTM PAL 2013.23.1. Scal bar is for Fig. 2D, other skulls are in comparative scale.

3 (version 3.08). The 3D reconstructions of the three studied skulls are in the video files
(Video S1–S6; https://zenodo.org/record/4117812#.X5FfUO28o2w).

Specimens were not allowed to be cut for histological purposes, thus micro-computed
tomography (microCT) imaging was used to investigate the cross-sectional structure of
selected cranial elements and their ornamentation. MicroCT scanning of fossil and recent
bones was conducted in the laboratory of the Carl Zeiss IMT Austria GmbH (Budaörs,
Hungary), using a Zeiss Metrotom computer tomograph with interslice distances of 130
μm. CT scans of the 14 fossil and extant specimens used in this study is available at
morphosource.org:

MDE R 43 (https://doi.org/10.17602/M2/M170133);
MDE R 20 (https://doi.org/10.17602/M2/M170132);
MDE R 47 (https://doi.org/10.17602/M2/M170147);
MDE R 45 (https://doi.org/10.17602/M2/M170135);
MTM PAL 2003.12 (https://doi.org/10.17602/M2/M170134);
MTM PAL 2007.27.1 (https://doi.org/10.17602/M2/M170137);
MTM PAL 2007.28.1 (https://doi.org/10.17602/M2/M170138);
MTM PAL 2010.1.1 (https://doi.org/10.17602/M2/M170139);
MTM PAL 2020.31.1 (https://doi.org/10.17602/M2/M170146);
MTM PAL 2020.31.1 (https://doi.org/10.17602/M2/M170148);

MTM PAL 2020.32.1 (https://doi.org/10.17602/M2/M170142);
MTM PAL 2020.32.1 (https://doi.org/10.17602/M2/M170145);
MTM PAL 2020.33.1 (https://doi.org/10.17602/M2/M170143);
MTM PAL 2020.34.1 (https://doi.org/10.17602/M2/M170144).

## RESULTS

As revealed by microCT images of extant lizards, the presence of osteoderms across the skull is often associated with a thin radiolucent or unossified seam separating the overlying ornamentation from the underlying cranial element (e.g., Figs. 1A and 1C). In contrast, among species that develop their ornamentation by the exaggerated outgrowth of individual elements (and not the coossification of osteoderms), this radiolucent seam is absent (Figs. 1E and 1G). Although the superficial layer of bone is typically invested with many small openings and canals and that pass into the cancellous core (Figs. 1C2 and 1D2), the microCT data does not reveal an obvious boundary between cranial ornamentation and the underlying compact cortex.

### Cranial ornamentation in *Hungarosaurus*
#### *Premaxilla*

Premaxillae are preserved in four specimens, including two isolated elements along with the holotype skull (MTM 2007.26.1.-2007.26.34.) and in MTM PAL 2020.31.1. (Figs. 3A–3D). The smallest premaxilla (MTM V.2003.12) is almost half the size of the holotype (Fig. 3A), and thus likely represents a juvenile or subadult individual (*Ősi & Makádi, 2009*). Premaxillae are unfused to each other in all specimens. Ornamentation can be observed on all the specimens including the smallest element, but does not overlap the sutures between the two premaxillae, or the borders with the nasals and maxillae. On the smallest specimen (MTM V.2003.12), the ornamentation is formed by various deep, relatively large pits and grooves present both anteriorly and laterally reaching the premaxilla-maxilla contact. In addition, various nutritive foramina are present further suggesting the still active growth of this bone. This ornamentation is thickest along the anterior margin. On the larger specimens, the surface of the ornamentation is very slightly irregular, pitting is less extensive and various shallow holes (diameter 2–3 mm) are present (Figs. 3C and 3D). Ornamentation in the larger specimens is restricted to the anterolateral and ventrolateral magins of the premaxilla (Fig. 3D) and composed of irregularly shaped, 1–3 mm thick, flat bumbs with branching morphology. Pits and grooves are less extensive but wider compared to those on the smaller premaxilla. MicroCT scanning of the three smallest premaxillae (Figs. 3A–3C) indicates that there is no evidence of a seam of separation between the superficial cranial ornamentation and the underlying cranial element, similarly to that seen in extant lizards (Figs. 1C2 and 1D2).

#### *Nasal*

Nasals (Figs. 3E–3H) are preserved for the skulls of MTM PAL 2020.31.1., MTM PAL 2020.32.1. and the holotype (*Ősi et al., 2019*, Figs. 3E, 3F, 3H, Data S1 and Video S1–S6). There is also an isolated, complete right nasal (MTM PAL 2020.34.1., Fig. 3G).

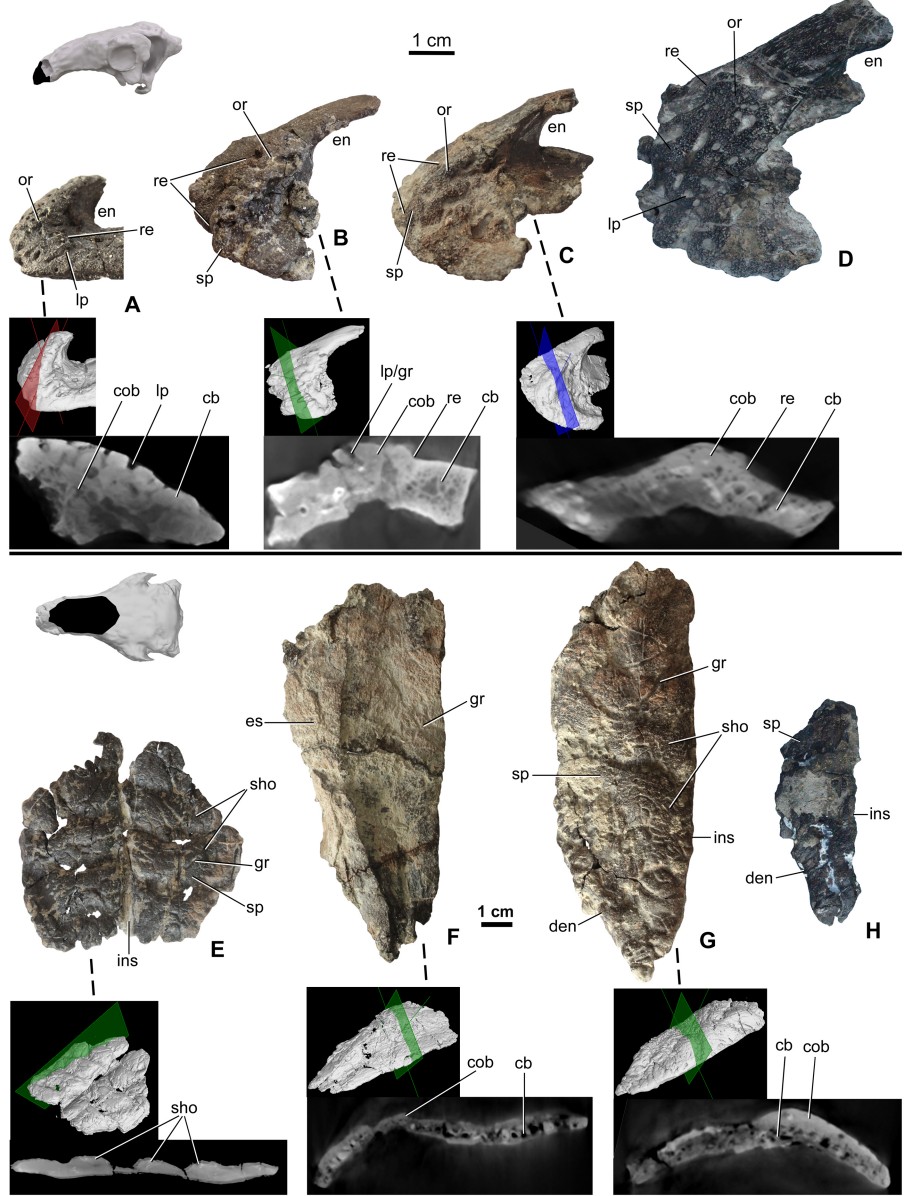

**Figure 3 Ontogenetic change of the cranial ornamentation on the premaxillae (A–D) and nasals (E–H) of *Hungarosaurus*.** Each element is visualized in surface view, three-dimensional surface rendering of microCT images, and microCT cross-sectional view. (A) Right premaxilla of MTM V 2003.12. (mirrored) in left lateral view. (B) left premaxilla of MTM PAL 2020.33.1. in left lateral view. (C) Premaxilla of MTM PAL 2020.31.1. in left lateral view. (D) Holotype premaxilla in left lateral view. (E) Nasals of MTM PAL 2020.32.1. in dorsal view. (F) right nasal from MTM PAL 2020.31.1. (G) Right nasal (MTM PAL 2020.34.1.) in doral view. (H) Holotype nasal fragment (mirrored). Abbreviations: cb, cancellous bone; cob, compact bone; den, dorsal margin of external nares; en, external nares; es, eroded surface; gr, groove; ins, internasal suture; lp, large pits; or, ornamentation; re, raised edge; sho, ornamentation in shingled arrangement; sp, small pits.

Similar to the premaxillae, nasals are unfused, a feature that is characteristic of skeletally immature ankylosaurs (e.g., *Pinacosaurus*, ZPAL MgD-II, (*Maryańska, 1977*); the holotype skull of *Europelta*, (*Kirkland et al., 2013*) and *Kunbarrasaurus* (*Molnar, 1996*; *Leahey et al., 2015*), but otherwise uncommon to ankylosaurs. Despite evidence of weathering, ornamentation is present along the dorsal surface of all the nasals (Fig. 3F). As revealed by MTM PAL 2020.32.1, the cranial ornamentation across the nasal consists of four or five transversely oriented and weakly shingled hummocky ridges (Fig. 3E). A comparable, hummocky-shingled ornamentation is also observed on the nasals of *Pawpawsaurus* (*Lee, 1996*; *Paulina-Carabajal, Lee & Jacobs, 2016*). Although hummocky ornamentation is also preserved on the larger specimen (MTM PAL 2020.34.1., Fig. 3G), the shingled arrangement is less obvious. Cranial ornamentation across the nasal is further characterized by a network of small pits (diameter: 0.5–3 mm) and grooves (length: 5-20 mm). None of the ornamentation across the nasal reaches the premaxilla-nasal, internasal and maxilla-nasal sutural borders on any of the studied specimens. Along the maxillary and prefrontal sutural borders, the nasal becomes thinner and the ornamentation abruptly ends, resulting in an irregular, step-like raised edge towards the maxilla and prefrontal. The nasal connects to the frontal via a scarf joint and, unlike the other sutural contacts, the pattern of cranial ornamentation appears to overlap the nasal process of the frontal (Data S1).

MicroCT scans from the nasals of three different individuals (MTM PAL 2020.32.1., MTM PAL 2020.31.1., MTM PAL 2020.34.1.) revealed no indication that cranial ornamentation was separated from the nasal in any of the specimens. Instead, the nasal (including cranial ornamentation) reveals a diploë organization, with a thicker layer of compact bone along the external (dorsal) surface as compared with the cancellous internal (ventral) surface (Figs. 3E–3G).

*Prefrontal-supraorbital-frontal complex*
The skull roof between the orbits is partly preserved from a number of specimens (Table 2), including MTM PAL 2020.32.1., MTM 2007.27.1 (an isolated left frontal), and MTM PAL 2013.23.1 (Figs. 4A–4C). In all specimens, the cranial elements posterior to the nasals (i.e., the temporal region of *Vickaryous & Russell, 2003*) are completely fused and their sutural boundaries obliterated. Cranial ornamentation on MTM PAL 2020.32.1. (Fig. 4A, Fig. S2, Data S1 and Video S3–S6) includes a number of large, deep pits (diameter: 2–4 mm) and relatively short, shallow grooves. These grooves appear to radiate from a near-central domed area, corresponding to the position of the parietals. Similar to the nasals, the surface of these elements is further ornamented by very small pits (diameter: 0.2–1 mm) and grooves (length: 1–5 mm). The isolated frontal (Fig. 4B) is ornamented by various small, deep pits (diameter: 1–3 mm) and grooves (width: 1–3 mm). Similar to the nasals, microCT scans of the frontals revealed diploë structure, with a thicker layer of compact bone along the external (dorsal) surface, and no radiolucent seam between cranial ornamentation and the element proper. Some pits pass through the compact bone into the deeper cancellous bone whereas some wider holes (diameter: 2–3 mm) and channels of the cancellous part enter and end into the upper compact bone.

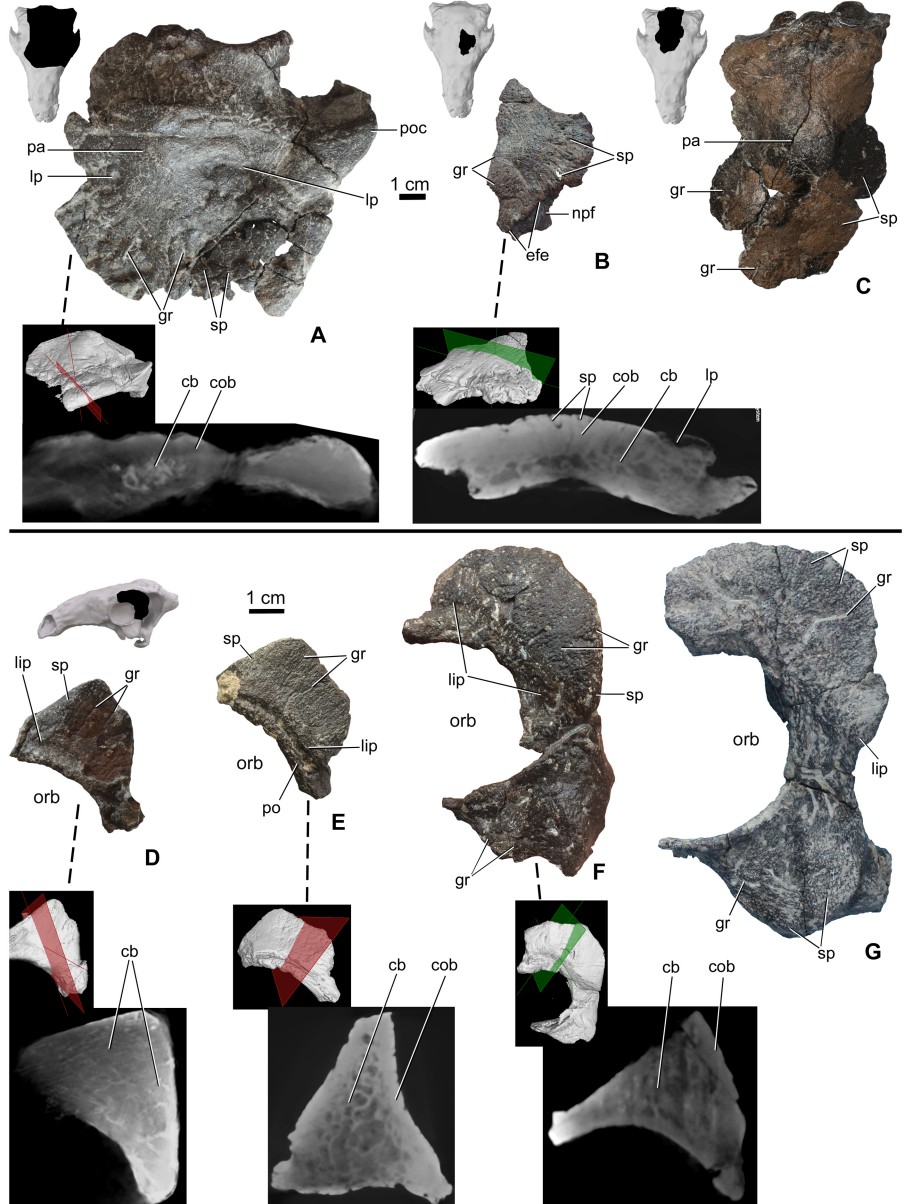

**Figure 4** **Ontogenetic change of the cranial ornamentation on the skull roof and orbital region of** *Hungarosaurus.* Each element is visualized in surface view, three-dimensional surface rendering of microCT images, and microCT cross-sectional view. (A) Skull roof of MTM PAL 2020.32.1. in dorsal view. (B) MTM 2007.27.1. left fragmentary frontal in dorsal view. (C) MTM PAL 2013.23.1. basicranium and partial sull roof in dorsal view. (D) Postorbital of MTM PAL 2020.32.1. (E) MTM 2007.28.1. left postorbital. (F) MTM 2010.1.1. left postorbital and jugal. (G) Holotype postorbital and jugal (mirrored). Abbreviations: cb, cancellous bone; cob, compact bone; efe, edge of frontal elaboration; gr, groove; lip, depressed "lip" at transition to softer skin; lp, large pits; npf, nasal process of frontal; orb, orbit; pa, parietal; po, postorbital; poc, postorbital crest; sp, small pits.

*Postorbital-jugal*

Portions of the postorbital and jugal are preserved that represent a number of different size classes (and presumably ontogenetic stages), including MTM PAL 2020.32.1. (Fig. 4D), two isolated specimens, MTM 2007.28.1. (Fig. 4E) and MTM 2010.1.1. (Fig. 4F), and the holotype (Fig. 4G, Data S1 and Video S1–S6). Characteristically, the long axis of the postorbital of *Hungarosaurus* passes along the posterodorsal margin of the orbit with a variably projecting crest-like caputegulum. In the smallest referred specimens (MTM PAL 2020.32.1., MTM 2007.28.1., Figs. 4D and 4E), this crest has a dorsoventral height/anterodorsal-posteroventral length ratio of 0.58, whereas in the larger specimens this ratio is reduced to 0.5–0.45 (MTM 2010.1.1., holotype, Figs. 4F and 4G). As a result, the crests in the larger specimens encircle more of the orbit, both dorsally and caudally (i.e., towards the jugal process). In addition, the crests of the smaller specimens are more rugose than the larger specimens, and are ornamented by a larger number of small, deeply opening pits and/or neurovascular canals. In contrast, these canals are largely absent from the largest specimen. As for the other cranial elements, microCT scans reveal no evidence of separation between the cranial ornamentation and the underlying elements (Figs. 4D–4F), with a similar pattern of compact bone surrounding a cancellous core.

The posteroventral margin of the orbit receives contributions from the jugal (and possibly the quadratojugal). In *Hungarosaurus*, the jugal is preserved in the holotype and by an isolated element (MTM 2010.1.1., Figs. 4F and 4G). The isolated specimen includes a relatively small quadratojugal boss with a short, ventrally pointed process, whereas that of the holotype is more rounded. As revealed by microCT scans, quadratojugal bosses are not separate elements from the quadratojugal. In all specimens, the surfaces are ornamented with rugose bone, including short neurovascular grooves (1–8 mm long) and small pits (0.3–1 mm). Similar to the postorbital crests, the smaller specimens are more heavily ornamented than the larger specimens. MicroCT scans of the jugal ornamentation (not figured) reveal a similar cross-sectional structure to the other skull elements, viz. a compact cortex surroudning a cancellous core.

*Parietal*

The area of the skull roof corresponding with the parietal is preserved in MTM PAL 2020.32.1. (Fig. 4A, Data S1 and Video S1–S6) and MTM PAL 2013.23.1. (Fig. 4C). This area forms a domed or vaulted complex, and most of its dorsal surface is relatively smooth or ornamented by shallow, short grooves and small pits (0.5-1 mm) on both specimens. On MTM PAL 2020.32.1., comparatively deep and wide grooves (> 5 mm) and large pits appear to roughly correspond with the positions of contact with the frontal, supraorbital and postorbital bones. MTM PAL 2013.23.1. (Fig. 4C) is at least 1.5 times larger than MTM PAL 2020.32.1., and thus most probably representing different ontogenetic stages. Based on microCT imaging (Fig. 4A), there is no evidence that osteoderms contribute to the development of cranial oramentation on this element.

## DISCUSSION

Cranial ornamentation is a hallmark feature of ankylosaurs (*Coombs Jr, 1978*; *Vickaryous, Maryańska & Weishampel, 2004*), and an emerging source of phylogenetic information (e.g., *Arbour & Currie, 2013*; *Arbour & Currie, 2016*). Although the skeletally mature pattern of cranial ornamentation may take the form of a series of variably shaped and sized polygons (referred to as caputegulae; *Blows, 2001*; see also *Arbour & Currie, 2013*), in some species these discrete features are not present. Regardless of the pattern formed, cranial ornamentation appears to form as a result of two potentially congruent processes: the coossification of overlying osteoderms with the skull, and the exaggerated outgrowth of individual cranial elements (*Vickaryous, Russell & Currie, 2001*; *Hill, Witmer & Norell, 2003*). The cranial material described here provides a rare opportunity to investigate the contribution of each of these processes in a European nodosaurid.

Using size as a proxy for age, we interpret the described specimens as representing a partial ontogenetic series of *Hungarosaurus* (Figs. 2, 3 and 4). The smallest specimen (MTM V.2003.12; estimated total skull length ∼15–17 cm) is approximately half the size of the largest (the holotype and MTM PAL 2013.23.1; estimated total skull length ∼34–36 cm). A fourth skull (MTM PAL 2020.32.1.; estimated total skull length ∼25 cm), is intermediate in size. Our findings reveal that cranial ornamentation, in the form of rugose texturing across the premaxilla and nasal, as well a sharp crest-like ridge along the postorbital, is already present in the smallest (= ontogenetically youngest) individuals examined. Although the pattern of cranial ornamentation changes as the individual gets larger, we found no evidence for the fusion or coossification of osteoderms with the underlying skull.

In *Hungarosaurus*, the smallest (= ontogenetically youngest) specimens have a more well-defined pattern of cranial ornamentation compared to larger (and presumably older) specimens. For example, the premaxilla of the smallest specimen has a more deeply pitted rugosity profile when compared to the larger specimens. Similarly, the pattern of small pits and grooves across the prefrontal-supraorbital-frontal complex and the parietal is more obvious on the smallest specimen. And while the nasal bone also demonstrates a well-developed pattern of transversely oriented pattern of hummocky rugosity, in smaller specimens these features form an anteriorly imbricated or shingle-like arrangement. In larger individuals the hummocky rugosity pattern is retained, albeit with a reduced (i.e., more shallow) profile. Whether this reflects an alternation in growth and maintenance of cranial ornamentation or the overlying keratinous skin structures, or evidence of sexual dimorphism or other form of signaling or identifier, remains unclear.

One of the most characteristic features of *Hungarosaurus* is the formation of a well-defined crest-like caputegulum on the postorbital. This structure is present in the smallest specimens (MTM PAL 2020.32.1., MTM 2007.28.1, Figs. 4D–4G), suggesting that it develops relative early during ontogeny, similar to the supraorbital horns of ceratopsians (*Horner & Goodwin, 2006*). MicroCT images reveal no evidence that this crest is formed by the coossification of an osteoderm with the postorbital. As for other features of cranial ornamentation, the morphology of the postorbital crest changes during ontogeny. In the smallest specimens, the shape of the postorbital crest is more acute compared with larger

(and presumably ontogenetically older) individuals. Near the margin of the orbit, the postorbital demonstrates a pronounced basal sulcus or 'lip' (sensu *Hieronymus et al., 2009*, Figs. 4D–4G). Although this feature was previously characterized as a fused osteoderm (*Ősi et al., 2012*), it is reinterpreted here, according to *Hieronymus et al. (2009)*, as evidence for a cornified sheath. A similar, well demarcated basal sulcus on the postorbital has also been reported for *Euoplocephalus* (*Vickaryous, Russell & Currie, 2001*).

Osteoderms do not contribute to the cranial ornamentation across the skull of *Hungarosaurus*. Our microCT data does not reveal any evidence that the cranial elements received a separate superifical contribution of bone, and there are no signs of osteoderms superimposed across sutural boundaries. Consequently, we predict that cranial ornamentation in *Hungarosaurus*, similar to the basal ankylosaur *Cedarpelta*, is the result of elaborated (exaggerated or exostotic) outgrowth of individual cranial elements. This elaborated/outgrowth form of cranial ornamentation has also reported for non-eurypodan thyreophorans such as *Scelidosaurus* and *Emausaurus* (*Norman, 2020*), as well as many extant lizard species (e.g., *Etheridge & De Queiroz, 1988*; *Powell et al., 2017*).

Similar to other ankylosaurs (e.g., *Arbour & Currie, 2013*), the pattern of cranial ornamentaion varies between specimens of *Hungarosaurs*. Although this variation is primarily interepreted as ontogenetic, the potential role of sexual dimorphism, individual differences, and taphonomic processes (e.g., deformation, weathering) cannot be excluded. Sexually dimorphic differences in cranial ornamentation has been suggested for a number of fossil archosaurs, including pterosaurs (*Bennett, 1992*; *Bennett, 2001*; *Naish & Martill, 2003*), ceratopsian dinosaurs (*Lehman, 1990*; *Sampson, Ryan & Tanke, 1997*; *Knell & Sampson, 2011*; *Borkovic, 2013*; *Hone & Naish, 2013*) and the ankylosaurid *Pinacosaurus* (*Godefroit et al., 1999*). With rare exceptions however, the limited number of specimens and/or incomplete preservation of skull material makes the identification of sex-related differences challenging (but see *Bennett, 1992*). Among the elements described herein, we did observe differences in size and shape. Although none of this variation cannot be separated from changes as a result of ontogeny, their potential use as dimorphic signals cannot be ruled out. For example, the postorbital crest of the holotype and MTM V 2010.1.1. encircles more of the orbit (dorsally and caudally), than those of MTM PAL 2020.32.1. and MTM 2007.28.1. (Figs. 4D–4G). Though the latter specimens are from smaller individuals, it remains possible that some of the morphological differences may also be related to dimorphism. Evidence from both fossil and extant species have made it clear that cranial ornamentation is often variable, and that the exclusive use of these features for taxonomic characterization should be viewed with caution (*Godefroit et al., 1999*; *Martill & Naish, 2006*). Future work on the cranial ornamentation of recent forms may bring us closer to the understanding of the cranial ornamentation of fossil taxa as well.

## CONCLUSIONS

The Santonian nodosaurid *Hungarosaurus* is represented by multiple individuals, including a partial ontogenetic series. Unlike some Late Cretaceous ankylosaurids, osseous

ornamentation in *Hungarosaurus* is restricted to individual elements, and does not appear to include the incorporation of osteoderms. In *Hungarosaurus*, cranial ornamentation was already well-formed in the smallest (= youngest) individuals. Although ontogeny appears to be a key source of variation, the contribution of individual differences, sexual dimorphism and even taphonomic processes cannot be ruled out.

**Institutional abbreviations**

| | |
|---|---|
| **IGM** | Institute of Geology, Ulaan Baatar, Mongolia |
| **MDE** | Collection of the Hungarian Dinosaur Expedition, Budapest, Hungary |
| **MTM** | Hungarian Natural History Museum, Budapest, Hungary |
| **TMP** | Royal Tyrrell Museum, Drumheller, Canada |

## ACKNOWLEDGEMENTS

We thank the reviewers Lucy Leahey, James Kirkland and Jelle Wiersma to their constructive comments that highy improved the manuscript. We are grateful to Doug Boyer for his generous help in depositing the CT data in Morphosource. We thank the 2000–2019 field crew for their assistance in the Iharkút fieldwork.

### Funding

This work was supported by the MTA-ELTE Lendület Dinosaur Research Group (No. 95102), National Research, Development and Innovation Office, Hungary (No. 116665, 131597), and the National Geographic Society (No. 7228-02, 7508-03). The funders had no role in study design, data collection and analysis, decision to publish, or preparation of the manuscript.

### Grant Disclosures

The following grant information was disclosed by the authors:
MTA-ELTE Lendület Dinosaur Research Group: 95102.
National Research, Development and Innovation Office, Hungary: 116665, 131597.
National Geographic Society: 7228-02, 7508-03.

### Competing Interests

The authors declare there are no competing interests.

### Author Contributions

- Attila Ősi conceived and designed the experiments, performed the experiments, analyzed the data, prepared figures and/or tables, authored or reviewed drafts of the paper, and approved the final draft.
- János Magyar conceived and designed the experiments, performed the experiments, prepared figures and/or tables, authored or reviewed drafts of the paper, and approved the final draft.

- Károly Rosta conceived and designed the experiments, performed the experiments, prepared figures and/or tables, and approved the final draft.
- Matthew Vickaryous performed the experiments, analyzed the data, authored or reviewed drafts of the paper, and approved the final draft.

## Data Availability

The osteological description of two skulls of *Hungarosaurus tormai* is available as a Supplemental File.

Six video files are available at Zenodo:

Ősi, Attila, Magyar, János, Rosta, Károly, & Vickaryous, Matthew. (2021). Supplementary video files for the Ősi et al. paper. Zenodo. http://doi.org/10.5281/zenodo.4117812.

CT scans of the 14 fossil and extant specimens used in this study are available at Morphosource:

MDE R 43 (DOI: 10.17602/M2/M170133);

MDE R 20 (DOI: 10.17602/M2/M170132);

MDE R 47 (DOI: 10.17602/M2/M170147);

MDE R 45 (DOI: 10.17602/M2/M170135);

MTM PAL 2003.12 (DOI: 10.17602/M2/M170134);

MTM PAL 2007.27.1 (DOI: 10.17602/M2/M170137);

MTM PAL 2007.28.1 (DOI: 10.17602/M2/M170138);

MTM PAL 2010.1.1 (DOI: 10.17602/M2/M170139);

MTM PAL 2020.31.1 (DOI: 10.17602/M2/M170146);

MTM PAL 2020.31.1 (DOI: 10.17602/M2/M170148);

MTM PAL 2020.32.1 (DOI: 10.17602/M2/M170142);

MTM PAL 2020.32.1 (DOI: 10.17602/M2/M170145);

MTM PAL 2020.33.1 (DOI: 10.17602/M2/M170143);

MTM PAL 2020.34.1 (DOI: 10.17602/M2/M170144).

## Supplemental Information

Supplemental information for this article can be found online at http://dx.doi.org/10.7717/peerj.11010#supplemental-information.

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
