# Peer review of "Cranial ornamentation in the Late Cretaceous nodosaurid ankylosaur Hungarosaurus"

_PeerJ, doi:10.7717/peerj.11010_

## Round 0.1 · original submission · Minor Revisions

The reviewers are overall quite enthusiastic about the manuscript (as am I). They have suggested a number of relatively minor revisions, which should either be incorporated into the revised manuscript or rebutted in the accompanying resubmission letter.

·

Basic reporting

An excellent description of the cranial elements of Hungarosaurus.

The only glaring weakness is the lack of comparisons with other European nodosaurids such as Europelta and Struthiosaurus for which cranial material is known.

Figures are well done.

The supplemental rotating skull models are exceptionally informative.

It should be noted that the holotype skull of Europelta also has unfused nasals (Kirkland et al., 2013),

Experimental design

Wonderful no comment!

Validity of the findings

In full agreement the conclusions regarding the origin of the cranial ornamentation.

·

Basic reporting

The manuscript is clear and unambiguous. Most issues with the manuscript were grammatical, particularly with sentence structure and the correct use of verbs/pronouns etc.
Literature review is thorough, however, a couple of issues arose – see suggestions below and in text. Tables of data are excellent and thorough.
Structure conforms to industry norms
Figures are also good, although I have made some suggestions below.

Experimental design

Original research in terms of the group nodosaurids, but based on previous work for the sister group ankylosaurids. Nice to see it finally happen!
Research question is well defined and relevant. ‘Gap’ in the literature answered. Would like to see mention of future work for other nodosaurids. Investigation was thorough and methods sufficiently detailed.

Validity of the findings

All data has been provided, and arguments and inferred conclusions are sound. I would like to see a sentence or two about potential future work and the what the conclusions infer for nodosaurids in general.

Additional comments

Corrections/Suggestions
Main text
I have also made minor corrections/suggestions within the text document – see the manuscript document in conjunction with comments below

no need for a comma after using – e.g. see lines 44, 53, 54, 61 etc.
no need for a comma after using – i.e. see line 75

Line 48-49: Here you list ‘cranial elaboration’ first, and ‘fusion of osteoderms’ second, however in the main text your information and arguments discuss ‘fusion of osteoderms’ first and ‘cranial elaboration’ second. To maintain flow throughout the text, consider swapping how you first list the options.

Line 63: (as well as throughout text and in reference list) Maryańska not Maryanska

Line 73: Molnar (1996) provides a description only of the cranial ornamentation. He goes into further detail in Molnar (2001 – within Carpenter’s The Armoured Dinosaurs). In either texts, he does not discuss the ontogenetic origin of the cranial ornamentation. It is discussed in Leahey et al. (2015) Cranial osteology of Kunbarrasaurus ieversi (see pgs 36-7). Furthermore I would not consider Kunbarrasaurus an ankylosaurid, the most recent published phylogenetic analyses and my own work on it place it as a basal ankylosaurian.

Line 75-9: Sentence is very convoluted, consider revising to:
Based on the investigation of multiple specimens of the ankylosaurids Euoplocephalus and Pinacosaurus, including material attributed to subadult (i.e., not skeletally mature) individuals, the cranial ornamentation of these forms are interpreted involving both the coosification of osteoderms with the skull and the exaggerated outgrowth of individual cranial elements (Vickaryous, Russell & Currie, 2001; Hill, Witmer & Norell, 2003; although see Carpenter et al., 2001).
* This is also the process of cranial ornamentation in Kunbarrasaurus ieversi (Leahey et al 2015)
Line 80: Basal ankylosaurian Kunbarrasaurus ieversi (Leahey et al 2015) has both osteoderms and cranial elaboration

Line 82-4: I’m not sure if this is the best example of a nodosaurid to discuss the group in general. The nodosaurid noted (SMU 7244 – Jacobs et al, 1994). Is a VERY juvenile specimen, and thus osteoderms may not yet have formed (i.e. similar to extant crocs – osteoderms do not develop very early). Also, the specimen is very fragmentary, and osteoderms if present may not have preserved. I think that this needs mentioning at least.

Line 90: I think here would be a good place to introduce your use of extant forms to help in your understanding of cranial ornamentation. I was a bit confused when you went straight into extnant forms from ankylosaurs. Just a sentence or 2 is needed. - e.g. Herein we investigate and utilize extant forms of sauropsids to better understand the processes of ontogenetic development in ankylosaurians.

Line 111: maybe add...Thus fusion of osteoderms to the underlying cranial elements is species-specific process????

Line 112: it is unclear to which skulls you are referring to

Line 175-6: Consider rephrasing:
…..from that of the holotype of Hungarosaurus. Herein we interpret this difference as either a factor of ontogeny or sexual dimorphism (see discussion below).

Line 230: longitudinal is an unclear direction, use anatomical directions e.g. lateral. Also, here you use ‘profile’ but you have used ‘view’ throughout you text, best to be consistent throughout

Line 238-41: It is unclear to which specimen you are referring to here, and similarly in the next paragraph (Lines 242-246), always good to use the specimen numbers to be ultra clear

Line 258: grooves are they 1-5mm wide? Long?

Line 263: do these channels reach the surface or end within the compact bone?

Line 278-9: Consider revising, to make clear:
The postorbital elements that underwent microCT (Fig. 4D-F), all reveal cancellous bone…..

Line 280-2: Consider swapping the sequence of the first two sentences – it will help with flow

Line 338: possibly add at the end of the sentence: (discussed further below)

Line 343-4: “co-osification of multiple elements” I’m a little confused by this term. Are you referring to the coossification of dermal elements or cranial elements? i.e. for the latter, as is the case for Pinacosaurus grangeri; multiple supraorbitals
Need to define which elements are coosifying: dermal or cranial elements

Line 349: At no point in this manuscript or in the supplement can I find any reference/description/discussion on ‘cornified sheath’. This either needs to be expanded upon as to why you interpret this or remove from the manuscript.

Line 359: ‘non-ankylosaur ankylosauromorphs’ I would hesitant in using this term as it has yet to be confirmed with recent phylogenetics. And the phylogenetics that Norman refers to ‘Carpenter 2001’ is consider unconventional. Probably safer using the term ‘non-eurypodan thyreophoran’
Also, the reference should be Norman 2020: the postcranial description NOT the cranial description as listed here and in the reference list.

Line 336-362: This form of cranial ornamentation also occurs in the basal ankylosaurian Kunbarrasaurus ieversi (Leahey et al., 2015)

Line 411: It would be great to see a sentence or two about potential futurework and the what the conclusions infer for nodosaurids and ankylosaurians in general.

Figures
General comments to consider
Place scale bar in areas where it is not so busy
Some of the microCT images are a bit fuzzy, consider increasing the black/white contrast to exaggerate the differences in fusion
Figure 1:
1. This image was a little hard to follow. When you have multiple groups or related images within one figure consider using ‘Ai, ii, iii; Bi, ii iii’ instead of A, B, C, D etc. Then the reader knows the relationships without having to search through the figure caption multiple times
2. For the focused/magnified images B, D, F & H. Consider placing a small box shape on the corresponding whole cranium photo (A, C, E, G) to show the area of magnification OR label the corresponding bones on the whole cranium photo that are labelled in the magnified image.

Figure 2: put a scale in for the basicranium only

Figure 4: (line 780) depressed “lip”

Supplementary data notes
1. It is unclear to which fused sutural contacts you are referring to: pmx-pmx, nas-nas, or pmx-nas?
2. ‘shows some of’ - as your previous sentence states otherwise.
3. ‘large’ is an ambiguous term, large relative to what?
4. Add ‘with an’
5. consider revising to: All these are diagnostic features of Hungarosaurus (ref), and are present on MTM PAL 2020.31.1.
6. need to label on figure
7. Why does the blood vessel only supply ornamentation? If it is an ontogenetic feature than the bone element itself would be growing as well (?).
8. This dimension is unclear.
9. Consider revising this sentence to: The outer surface is strongly weathered (and relatively thin, max. thickness of 4-6mm) and thus not overly informative, but …..
10. Are the sutures preserved in other specimens of Hungarosaurus?
11. Reconsider revising to:
The nasals are at least twice as anteroposteriorly long than wide (although anteriorly they are not completely preserved), dorsally highly ornamented (for details see main text), trapezoid elements. At the nasal-frontal contact, the nasals overlap the anterior process of the frontals (Suppl. Fig. 2A, B).
12. Needs to be labelled on figure
13. Reconsider revising to:
In dorsal view, the squamosal is laterally- posterolaterally oriented and bears no significant ornamentation. Only some rugose texture can be seen on its mediodorsal surface (Suppl. Fig. 2E), extending towards the exoccipital. This may have been served for the attachment of the dorsal neck muscles.
14. Reconsider revising to:
Based on the diagnostic characters of Hungarosaurus tormai; the postorbital bearing a high and anterodorsal–posteroventrally elongated crest, and the mandibular quadrate condyle having rhomboidal articular surface (Ősi et al. 2019), we herein assign MTM PAL 2020.32.1 to Hungarosaurus. However the very slight differences in the morphology of these characters in conjunction with the relatively large orbits, unfused postorbital bones and the relatively small size of the skull suggests a subadult ontogenetic stage. Thus herein MTM PAL 2020.32.1. is consider an ontogenetically immature form of Hungarosaurus tormai.

·

Basic reporting

Introduction:
The introduction section is rather disjointed and does not really ties the topics together with regard to nodosaur cranial ornamentation and the lack of data regarding cranial osteoderm ontogeny, sexual dimorphism and taxonomic diversity. The introduction steers away from the central topic of this manuscript, which focuses chiefly on the cranial ornamentation of Hungarosaurus. The authors briefly state that they describe fragmentary skulls and elements that potentially represent different ontogenetic stages or sexually dimorphic individuals, but the introduction fails to clearly mention what research questions or problems the authors are aiming to answer with these specimens in the broader (taxonomic) sense and why this topic is scientifically relevant to ankylosaurian dinosaurs, and nodosaurids in particular.
Also, the cranial ornamentation in ankylosaurian dinosaurs is discussed very generally and briefly in two short paragraphs and would benefit from being expanded upon. Instead, the focus shifts heavily towards more extant archosaur and lepidosaur taxa cranial osteoderm morphology, ontogeny, and diversity. I understand what the authors are attempting to demonstrate with this comparative approach, but it loses the focus on ankylosauria, and nodosaurids in particular, entirely. The introduction would benefit from restructuring, focusing on the lack of cranial osteoderm ontogenetic and sexually dimorphic data across a wide range of ankylosaur taxa and how their data has the potential to elucidate some of these problems in extinct forms. Include the extant data because it is quite informative with regard to said topics, but it needs to be tied back together better to the central topic of the manuscript: Hungarosaurus and Ankylosauria more generally or nodosaurids more specifically. In its current state, the paragraphs on extant data are standalone and appear as separate sections that don't properly contextualize with Hungarosaurus.

Materials & Methods:
The authors utilized extant taxa as well for comparative purposes. These specimens need to be listed in this section as well, including the purpose of CT scanning (clearly for comparative analysis), equipment used and slice thickness for each specimen. State how many individuals of each species where scanned, where the specimens are from and housed, including specimen numbers.

Discussion:
The discussion section overall doesn't flow very well and does not really add much in terms of context or putting the data together. What does it all mean, and how can we utilize/implement the findings when continuing studies on ankylosaur cranial ornamentation? The discussion repeats largely what has already been mentioned in the Introduction section and thus feels repetitive and therefore redundant. The discussion needs to be re-written and focus clearly on the characteristics of these cranial features in the Hungarosaurus specimens, supported by comparisons made with the extant taxa. The authors have great CT data for both extant and extinct taxa and I think if you combine the similarities and differences (aided by an extra figure or two to highlight these difference/similarities) between extant taxa and Hungarosaurus, the authors can discuss the implications of these multiple Hungarosaurus specimens and cranial osteoderm variety in a much more focused sense.
Also, the authors emphasize dimorphism and cranial outgrowth in birds, osteoderm presence and coosification in squamates etc, based on CT data analysis. However, I would stress to also discuss and observe the possibility of interspecific variation in the taxa observed and discuss the possibility of this also (partially) being the case for Hungarosaurus. Phenotypic plasticity is very common in extant taxa, and has been well documented (especially in lizards, anoles, bats, etc), but it is surprisingly often overlooked in extinct taxa (even in cases where large numbers of specimens from a single population are available for study). I think it would be beneficial for the manuscript and the hypotheses stated to strengthen the arguments by:
1) including more extant specimens of the same species studied to observe the level of variation of osteoderms or bone outgrowth in adult males, females, males and females and juveniles. Compare for intraspecific and ontogenetic variation of cranial ornamentation and highlight the major differences/similarities.
2) compare these results with the CT data and qualitative data of Hungarosaurus ornamentation to better discuss the hypotheses of excessive outgrowth, potential sexual dimorphism and ontogeny in Hungarosaurus. This will result in a much stronger and more focused, to-the-point discussion.

Experimental design

No comment.

Validity of the findings

No comment.

Additional comments

First of all, I would like to thank the Editors for letting me review the manuscript titled: "Cranial ornamentation in the Late Cretaceous nodosaurid ankylosaur Hungarosaurus". I very much enjoyed reading the manuscript, and it is overall well written and provides important new data regarding the ontogenetic stages of cranial ornamentation for ankylosaurian dinosaurs; a topic that is in continuous need of study for this generally poorly understood group of dinosaurs.
I really like the figures; they are very clearly illustrated and professional. I made a few minor suggestions in the pdf but overall, they are effective and well done.
There are a few issues in this manuscript that should be addressed before the paper can be accepted for publication, and I highlighted these in the Basic Reporting section. Additionally, I provided an annotated pdf with minor comments, consisting mostly of typo corrections and some grammatical issues and some suggested revisions.
The paper has a lot of potential and is important for thyreophoran research, but in its current state it reads more like a descriptive paper rather than really focusing on the implications of the different states and stages of cranial ornamentation in the discussed Hungarosaurus specimens. It misses comparison with the extant CT data to support and contextualize the authors' arguments. This is unfortunate, because there is merit in this paper and the topic of ankylosaur cranial ontogeny and potential sexual dimorphism in cranial ornamentation is heavily understudied, making this manuscript a very much needed addition to this field of study.
I think the authors should think about adding a little more data (see suggestions above) and re-write the aforementioned sections to strengthen the arguments more convincingly. I would like to see these suggestions implemented and corrections addressed before the paper can be accepted for publication.
I am looking forward to read the published work.

---

## Round 0.2 · accepted · Accept

Thank you for your close attention to the comments from the reviewers. The manuscript is ready to move on to publication. I have identified a handful of very minor typos, which can be corrected after acceptance.